# Immunogenicity and Efficacy of TNX-1800, A Live Virus Recombinant Poxvirus Vaccine Candidate, against SARS-CoV-2 Challenge in Nonhuman Primates

**DOI:** 10.3390/vaccines11111682

**Published:** 2023-11-02

**Authors:** Mayanka Awasthi, Anthony Macaluso, Dawn Myscofski, Jon Prigge, Fusataka Koide, Ryan S. Noyce, Siobhan Fogarty, Helen Stillwell, Scott J. Goebel, Bruce Daugherty, Farooq Nasar, Sina Bavari, Seth Lederman

**Affiliations:** 1Tonix Pharmaceuticals, Frederick, MD 21701, USA; mayanka.awasthi@tonixpharma.com (M.A.); tonymac357@gmail.com (A.M.); dawn.myscofski@tonixpharma.com (D.M.); scott.goebel@tonixpharma.com (S.J.G.); farooq.nasar@tonixpharma.com (F.N.); sina.bavari@tonixpharma.com (S.B.); 2BioQual, Rockville, MD 20850, USA; jprigge@bioqual.com; 3Southern Research, Birmingham, AL 35205, USA; fkoide@southernresearch.org; 4Department of Medical Microbiology & Immunology, Li Ka Shing Institute of Virology, University of Alberta, Edmonton, AB T6G 2R3, Canada; noyce@ualberta.ca; 5Tonix Pharma Limited, A96 HW25 Dublin, Ireland; siobhan.fogarty@tonixpharma.com; 6Department of Microbiology, University of Pennsylvania Perelman School of Medicine, Philadelphia, PA 19104, USA; helen.stillwell@pennmedicine.upenn.edu; 7Tonix Pharmaceuticals, Chatham, NJ 07928, USA; bruce.daugherty@tonixpharma.com; 8Tonix Pharmaceuticals, Dartmouth, MA 02748, USA

**Keywords:** SARS-CoV-2, TNX-1800, vaccine, vaccine platform, horsepox virus, spike protein, antibody titer, immunogenicity

## Abstract

TNX-1800 is a synthetically derived live recombinant chimeric horsepox virus (rcHPXV) vaccine candidate expressing Wuhan SARS-CoV-2 spike (S) protein. The primary objective of this study was to evaluate the immunogenicity and efficacy of TNX-1800 in two nonhuman primate species challenged with USA-WA1/2020 SARS-CoV-2. TNX-1800 vaccination was well tolerated with no serious adverse events or significant changes in clinical parameters. A single dose of TNX-1800 generated humoral responses in African Green Monkeys and Cynomolgus Macaques, as measured by the total binding of anti-SARS-CoV-2 S IgG and neutralizing antibody titers against the USA-WA1/2020 strain. In addition, a single dose of TNX-1800 induced an interferon-gamma (IFN-γ)-mediated T-cell response in Cynomolgus Macaques. Following challenge with SARS-CoV-2, African Green and Cynomolgus Macaques exhibited rapid clearance of virus in the upper and lower respiratory tract. Future studies will assess the efficacy of TNX-1800 against newly emerging variants and demonstrate its safety in humans.

## 1. Introduction

Severe acute respiratory syndrome coronavirus 2 (SARS-CoV-2) is a member of the genus *Betacoronavirus* in the family *Coronaviridae*. The genome comprises of single-stranded, positive sense RNA, ~30 kb in length [1]. SARS-CoV-2 emerged from China in early 2020 to cause a global pandemic. To date, almost three-quarters of a billion human cases have been confirmed, with ~7 million deaths worldwide [2].

SARS-CoV-2 vaccines have proven to be cost-effective methods to prevent clinical disease, virus transmission, and reduce long-term healthcare burdens [3,4,5]. Consequently, as the SARS-CoV-2 infection continues to evolve [2], the urgency to develop vaccines against new variants that can generate robust and durable immunity remains a top priority for global public health [6].

Currently, several SARS-CoV-2 vaccines are authorized by the U.S. Food and Drug Administration (FDA), mRNA-based vaccines from Pfizer-BioNTech and Moderna, and protein subunit vaccines from Novavax [7,8,9,10]. A live attenuated adenovirus vaccine from J&J/Janssen COVID-19 vaccine [11] is no longer available for use in the United States as of 6 May 2023, due to the expiry of its authorization [12]. Although the mRNA and protein subunit vaccines can elicit humoral and cell-mediated responses, there are important drawbacks to the vaccines, such as requiring a two-dose regimen, a lack of durable immunity, and reliance on the cold chain.

In contrast to other vector platforms, poxvirus-based vaccines offer several advantages [13,14,15,16]. The vaccinia virus-based smallpox vaccine elicits robust, durable, and lifelong immunity following single-dose vaccination [17]. In addition, the recombinant modified vaccinia virus Ankara (MVA) vaccine expressing a prefusion-stabilized SARS-CoV-2 spike glycoprotein induces robust immunity and protection in mouse, Syrian hamster, and non-human primate models [18,19].

We recently developed a recombinant chimeric horsepox virus (rcHPXV)-based TNX-801 as a vaccine platform against the monkeypox virus (MPXV). The vaccine was well tolerated and immunogenic in multiple animal models, including non-human primates (NHPs) [20]. In addition, a single dose vaccination with TNX-801 was able to generate strong immunity and provided complete protection against lethal MPXV clade I challenge in NHPs [20]. Following successful NHP studies against MPXV, the TNX-801 platform was engineered to express the SARS-CoV-2 spike protein of the Wuhan strain (TNX-1800) [21]. TNX-1800 was well tolerated and immunogenic, and the vaccine didn’t cause any disseminated horsepox infection in Syrian golden hamsters and New Zealand white rabbits [21].

In this proof-of-concept report, we investigated the immunogenicity and efficacy of TNX-1800 in nonhuman primates, African Green Monkeys (AGMs), and Cynomolgus Macaques (CMs). A single dose of TNX-1800 vaccination was well tolerated and able to induce humoral and cell-mediated responses. Following the SARS CoV-2 challenge, the vaccine was able to significantly reduce virus replication/shedding in the respiratory tract.

## 2. Materials and Methods

### 2.1. Cells and Virus

Vero and BSC-40 cells were obtained from the American Type Culture Collection (ATCC, Manassas, VA, USA). Vero-E6 TMPRSS2 cells were obtained from the Vaccine Research Center-NIAID (Rockville, MD, USA). Cells were propagated at 37 °C in 5% CO_2_ in Dulbecco’s Minimal Essential Medium (DMEM) Invitrogen (Thermo Fisher Scientific, Waltham, MA, USA) containing 10% (*v*/*v*) fetal bovine serum (FBS) (Hyclone, (Logan, UT, USA), sodium pyruvate (1 mM) Invitrogen (Thermo Fisher Scientific, Waltham, MA, USA), 1% (*v*/*v*) non-essential amino acids Invitrogen (Thermo Fisher Scientific, Waltham, MA, USA), and 50 μg/mL gentamicin, Invitrogen (Thermo Fisher Scientific, Waltham, MA, USA). 

SARS-CoV-2 (WA 2020) isolate was obtained from Biodefense and Emerging Infections Research Resources Repository resources (BEI). TNX-1800 and SARS-CoV-2 stocks were generated via amplification on BSC-40 and Vero-E6 TMPRSS2 cells, respectively.

### 2.2. Generation of TNX-1800

TNX-1800 was constructed by homologous recombination, inserting the expression cassette encoding the SARS-CoV-2 Spike S gene, codon-optimized for vaccinia virus, into TNX-801 at the Δ200 gene locus. The expression cassette comprises the SARS-CoV2 Spike S gene that is operatively linked to a vaccinia virus early and late promoter (Pox E/L promoter) inserted upstream of the SARS-CoV-2 Spike S gene. It further comprises HPXV left and right flanking arms (Figure 1).

### 2.3. Ethics Statement

This work was supported by an approved Institutional Animal Care and Use Committee (IACUC) animal research protocol in compliance with the Animal Welfare Act, PHS policy, and other federal statutes and regulations relating to animals and experiments involving animals. The facility where this research was conducted is accredited by the Association for Assessment and Accreditation of Laboratory Animal Care (AAALAC International) and adheres to principles stated in the Guide for the Care and Use of Laboratory Animals, National Research Council, 2011 [22].

### 2.4. Study Design and Immunization Procedure

A total of eight African Green Monkeys (AGMs) and eight Cynomolgus Macaques of Mauritius origin (CMs) were purchased from vendors Primera Science Center (LaBelle, FL, USA) and PrimGen (Hines, IL, USA), respectively. Animals were subsequently quarantined in the ABSL-2 for a minimum of 42 days. Prior to release from quarantine, the animals were screened and were found to be negative for simian immunodeficiency virus (SIV), Simian Retrovirus (SRV)-1/2/3/4/5, simian T-lymphotropic viruses (STLV)-1, Measles, and Tuberculosis (TB). The NHPs were vaccinated via percutaneous route at day 0, and following vaccination the animals were bled at various time points to determine immunogenicity. At day 41 post-vaccination, the NHPs were challenged with SARS-CoV-2 (WA 2020) isolate via the intranasal/intratracheal (IN/IT) routes. Samples for different assessments were collected on the days indicated in Figure 2A and Figure 3A.

For the AGM study, four naïve monkeys (two male and two female) were vaccinated with 6.5 log_10_ PFU of TNX-1800 via scarification at the interscapular region on day 0 (Figure 2B). Similarly, for the CM study, four adult monkeys (two male and two female) were vaccinated with 6.1 log_10_ PFU of TNX-1800 via scarification at the interscapular region on day 0 (Figure 3B). For the control groups of each study, four naïve monkeys (two male/two female) were injected with a sterile solution of diluent (Tris-HCl 10 mM; pH 8.0). Following vaccine administration, a bifurcated needle was used to scarify the area by penetrating the skin vertically approximately 15 times.

For the AGM study, oropharyngeal (OP) swabs and Transoral Tracheal Lavage (TOTL) were collected according to the schedule indicated in Figure 2A: days 35/36 (TOTL only), day 41 (OP only), and days 47/48, 53, and 60/61 for RT-qPCR (quantifying SARS-CoV-2 viral genomes). PBMCs were collected for cytokine quantification. Animals were euthanized on day 47 (6 days post-infection; 1/sex/group) or 60 (19 days post-infection; 1/sex/group). 

For the CM study, blood collection occurred on the days indicated in Figure 3A and was used for the measurement of binding and neutralizing antibodies through enzyme-linked immunosorbent assay (ELISA) and plaque reduction neutralization (PRNT) assay. PBMCs were also isolated for flow cytometry analysis and cytokine quantification. Nasal swabs and bronchoalveolar lavage (BALV) samples were collected on days 42 through 45 and days 42/45, respectively. All animals were euthanized either on day 48 or day 50.

### 2.5. SARS-CoV-2 Challenge and Back-Titration of Inoculum

The animals were anesthetized intramuscularly with 10 mg/kg of ketamine. SARS-CoV-2 virus stocks were diluted in sterile PBS to a dose level of 5 log_10_ PFU TCID_50_ per 2 mL volume. Syringes of 2 mL volume were prepared for each animal. The animals were challenged via the intranasal route and followed by the intratracheal route. For the intranasal route, 0.5 mL of the viral inoculum was administered dropwise into each nostril for a total of 1 mL volume per animal. Upon administration, the animal’s head was tilted back for ~20 s. For the intratracheal route, 1 mL of diluted virus was delivered intratracheally using a French rubber catheter/feeding tube, size 10, sterile (cut 4″–6″ in length). The syringe containing the inoculum was attached to a sterile French catheter or feeding tube. The small end of the feeding tube with the inoculum was inserted into the glottis. Once in place, the inoculum was injected into the trachea and then the catheter was removed from the trachea. Following inoculation, the animals were returned to respective housing units and monitored until fully recovered. New or sterilized equipment was used for each animal. The study animal was then inoculated via the intranasal route. The remaining inoculum was aliquoted into cryovial tubes and stored at −70 °C or below. The stored challenge inoculum was later back-titered via a TCID_50_ assay for verification of proper dose level.

### 2.6. Enzyme-Linked Immunosorbent Assay (ELISA)

A standard indirect ELISA was performed to analyze serum samples for binding antibodies to the SARS-CoV-2 spike protein. For this assay, 96-well plates were coated with 50 µL of SARS-CoV-2 spike protein (Sino Biological, Beijing, China) diluted to 2 µg/mL in carbonate-bicarbonate buffer (CBB, Sigma-Aldrich, St. Louis, MO, USA). Plates were incubated overnight at 2–8 °C. Unbound coating antigen in each well was removed by washing with PBS + 0.05% Tween-20 and then blocked with PBS + 1% BSA. Test and positive control samples were diluted in assay diluent (PBS-Tween 20; 1% BSA) to a starting point dilution of 1:20, followed by four-fold serial dilutions. Once blocking was completed, the blocking buffer was removed by inversion, plates were washed, and samples were added. Plates were incubated for 60 to 70 min at room temperature, followed by washing with PBS + 0.05% Tween-20 to remove unbound sera. Secondary detection antibody (Goat anti-Monkey IgG (H + L) Secondary Antibody, HRP, Invitrogen, Carlsbad, CA, USA) was added at a dilution of 1:10,000, and plates were incubated for 60 to 70 min at room temperature. Unbound antibodies were subsequently removed by washing with PBS + 0.05% Tween-20 and a final wash with PBS alone. To develop, 1-Step Ultra TMB substrate (SeraCare Life Sciences, Gaithersburg, MD, USA) was added to each well and the plate was developed. The reaction was stopped after 10 min to 15 min with TMB stop solution (SeraCare Life Sciences). The plates were read within 30 min at 450 nm using a Thermo Lab Systems Multiskan spectrophotometer.

### 2.7. Neutralization Assay

A SARS-CoV-2-specific plaque reduction neutralization titer (PRNT) assay was used to determine if the vaccinations had elicited neutralizing antibody responses in 6-well or 96-well formats. Serum samples were heat-inactivated for 30 min at 56 °C. Briefly, serum samples were serially diluted in media and added to equal volumes of a fixed dilution of the SARS-CoV-2 containing 2.0 log10 PFU. The serum–virus mixture was then incubated for 1 h at 37 °C. Subsequently, 250 µL of each serum–virus mixture was added in duplicate to a confluent monolayer of Vero cells and incubated at 37 ± 2 °C, 5% CO_2_ for 1 h. Following incubation, 1 mL of 0.5% methylcellulose in DMEM + 2% FBS was added to each well and incubated at 37 ± 2 °C, 5% CO_2_ for 3 days. The virus–cell mixture was incubated at 37 ± 2 °C; 5% CO_2_ for 48 ± 4 h, fixed with cold methanol, and then stained with 0.2% Crystal Violet solution for the enumeration of the plaques. Neutralization end-point titers were calculated and based on the reciprocal dilution of the test serum that produced 50% plaque reduction compared to the virus-only control (PRNT_50_).

### 2.8. Intracellular Cytokine Staining for SARS-CoV-2-Specific Immune Responses

Intracellular cytokine staining via flow cytometry was performed from cryopreserved PBMCs. Cells were thawed in media containing >5 U/mL of benzonase and resuspended in complete RPMI media supplemented with 10% FCS, L-Glutamine, and penicillin–streptomycin at a concentration of 1 × 10^7^ cells/mL. One million PBMCs per well were resuspended in 100 µL of R10 medium supplemented with CD49d, CD28 (Life Technologies), and CD107a (BioLegend) monoclonal antibodies. Each sample was tested with mock (100 uL of R10; background control), synthetic peptide pools spanning the SARS-CoV-2 spike protein (Peptivator S Peptide Miltenyi cat#130-126-700, Peptivator S1 Peptide Miltenyi cat#130-127-04 at a final concentration of approximately 1 µg/mL of each peptide) or 10 pg/mL phorbol myristate acetate (PMA), 1 µg/mL ionomycin (Sigma-Aldrich) (100 uL; positive control) and incubated at 37 °C for 1 h. After incubation, 1 µL/mL of GolgiPlug containing Brefeldin A (BD Biosciences, San Jose, CA, USA) was added to each well and incubated at 37 °C for 12 h. The next day, the cells were washed twice with DPBS, stained with live/dead Zombie Aqua dye for 15 min, and then stained with predetermined titers of monoclonal antibodies against CD3, CD8, and CD4 for 30 min. The cells were subsequently washed twice with 2% FBS in DPBS buffer and incubated for 15 min with 200 uL of BD Cytofix/CytoPerm Fixation/Permeabilization solution. The cells were washed twice with 1× Perm Wash buffer (BD Perm Wash Buffer 10× in the Cytofix/CytoPerm Fixation Permeabilization kit diluted with MilliQ water and passed through a 0.22 um filter) and stained intracellularly with monoclonal antibodies against IFN-γ and IL-4 for 30 min. The fixed cells were transferred to 96-well round bottom plates and analyzed by the Cytek Aurora flow cytometer (Cytek Biosciences, Freemont, CA, USA). Analysis of the acquired data was performed using FlowJo Software (Version 10, BD, Ashland, Oregon). Analysis was performed using FlowJo Software (Version 10.6.2, BD, Ashland, Oregon). Peptide-specific responses were calculated by subtraction of the unstimulated controls from the peptide-stimulated samples.

### 2.9. Collection of Oropharyngeal Swabs

For the oropharyngeal swab, the animal’s mouth was cleaned of excess saliva and food particles. The swab was inserted into the animal’s mouth, taking care not to touch any surface of the mouth until the tip of the swab was past the base of the tongue. The swab was pressed against the dorsal surface of the pharyngeal area, rolled across the surface, and removed carefully to avoid touching any other mouth surfaces. Following collection, the swabs were placed into a collection vial (2/specimen). The samples were snap-frozen immediately following collection and stored at or below −80 ± 10 °C until analyzed by RT-qPCR.

### 2.10. Collection of Transoral Tracheal and Bronchioalveolar Lavage

The Transoral Tracheal Lavage (TOTL) procedure was performed by inserting a tube into the trachea. Once the end of the tube was situated approximately at the mid-point of the trachea, a syringe containing up to 5 mL of sterile PBS was attached to the tube, and the medium was slowly instilled into the trachea. Once the instillation was complete, negative pressure was immediately applied via the same syringe to collect as much of the PBS as possible. The samples were snap-frozen immediately following collection and stored at or below −70 °C until analyzed.

The bronchoalveolar lavage (BAL) procedure was performed by the “chair method”. The NHP was placed in dorsal recumbency in the chair channel such that the handle was tightened into the position supporting the animal’s weight. A catheter tube was premeasured and marked before placement. The animal’s head was tilted back and down below the edge of the chair channel. The catheter tube was then inserted into the animal’s trachea via a laryngoscope during aspiration. A total of 10 mL was flushed through the tube. Upon sample completion, the handle at the base of the chair was loosened and the animal was removed from the chair. The volume instilled and recovered from each animal, as well as any presence of blood in the BAL samples, was recorded. The animal was monitored until fully recovered. The collected BAL samples were placed immediately onto wet ice and processed for the isolation of fluid. The sample was centrifuged to pellet debris, the supernatant was collected, and then it was aliquoted.

### 2.11. SARS-CoV-2 Viral Load by Quantitative PCR (RT-qPCR)

Viral load from the SARS-CoV-2 challenge was assessed using quantitative RT-qPCR for the detection of SARS-CoV-2 genomes in the total RNA extracted from the biospecimen using a QIAcube robot according to the manufacturer’s protocol. The samples collected as per study protocol, including Oropharyngeal (OP) swabs and transoral tracheal lavages (TOTL), were processed for total RNA extraction. The resulting RNA was eluted with nuclease-free H_2_O and stored at −70 °C or below prior to RT-qPCR testing. The SARS-CoV-2 viral Nucleocapsid (N) gene-specific- RTqPCR- duplex assay was designed based on using two independent PCR targets: the SARS-CoV-2 N-gene sequences and a synthetic RNA added to all control and test samples (Internal Amplification Control or IAC). The CDC-reported N-2 primer and probe set labeled with a specific fluorescent signal (FAM) were used to amplify SARS-CoV-2 RNA, releasing FAM bound to the intact N-2 probe during the target amplification. Xeno synthetic RNA-specific primers and probes were used to quantify the IAC reaction (ThermoFisher Scientific, VetMax Xeno Cat. # A29767). Purified single-stranded RNA (ssRNA) for the SARS-CoV-2 standard curve was generated by in vitro T7 transcription from a PCR fragment comprising the T7 promoter and SARS-CoV-2 (N) gene sequences. Synthetic T7 transcripts were quantified using the Quant-iT Ribogreen RNA assay kit. All RT-qPCR reactions were performed using the TaqMan Fast Virus 4× Master Mix. Values <LOD (2.01 copies/reaction) were considered negative, and values ≥LOD were considered positive. In PCR reactions where IAC was detected, RNA samples were diluted 4-10-fold and retested. The group results are reported as a Geometric mean (Gmean) genome copy number per mL or per swab.

### 2.12. SARS-CoV-2 Viral Load Quantified via 50% Median Tissue Culture Infectious Dose (TCID50) Infectious Viral Load Assay (TCID_50_)

Vero TMPRSS2 cells were seeded overnight at 25,000 cells/well at 37 °C, 5.0% CO_2_ to achieve 80–100% confluent. The medium was aspirated and replaced with 180 µL of 1X DMEM supplemented with 2% FBS and 10 ug/mL gentamicin. Twenty (20) µL of the sample was added to the top row in quadruplicate and mixed with a pipette. Using a pipette, 20 µL was transferred to the next row and repeated down the plate (columns A-H), representing 10-fold dilutions. The tips were disposed of for each row and repeated until the last row. Positive (virus stock) and negative (media) controls were included in each assay. The plates were incubated at 37 °C, 5.0% CO_2_ for 4 days. The cell monolayers were visually inspected for cytopathic effects (CPE). Uninfected wells had a clear confluent cell monolayer while infected wells displayed significant virus-induced cytopathic effects (CPE). Uninfected wells had a clear confluent cell monolayer while infected wells had little cell presence. The presence of CPE was marked on the lab form as a “+” and the absence of CPE as “0”. The TCID_50_ value was calculated using the Reed–Muench formula. 

### 2.13. Statistical Analysis

These studies were designed to achieve 80% power to determine whether the TNX-1800 is more efficacious than the control. All the statistical evaluations were performed using GraphPad Prism (GraphPad Software, v9, San Diego, CA, USA). The sample size was determined using the formula “10/*k* + 1”, where *k* = number of groups and *n* = number of subjects per group. The number of animals that were studied per group of Cynomolgus and African green macaques is indicated in the respective study designs. Statistical significance was determined using descriptive statistics and the ordinary one-way ANOVA followed by Turkey’s multiple comparison test unless otherwise stated. Values were considered significantly different at *p* ≤ 0.05 (*).

## 3. Results

### 3.1. African Green Monkeys (AGMs) Challenge Study

TNX-1800 elicits immune responses against SARS-CoV-2 in AGMs. TNX-1800 immunization was well tolerated in all monkeys and no serious adverse events were observed. All four animals vaccinated with TNX-1800 showed a measurable vaccine “take”, or cutaneous reaction, indicating successful vaccination, and all animals seroconverted as early as day 14 (Figure 4A and Appendix A). The magnitude of the neutralizing antibody response elicited as measured by geometric mean titers (GMTs) detected 14-, 21-, 28/29-, and 41-days post-vaccination ranged from 47–146.

On day 41, the animals were challenged with the SARS CoV-2 isolate WA1/2020 strain via the intranasal and intratracheal routes. Viral shedding in the upper respiratory tract (URT) was assessed by oral swabs (OP) and viral load in the lower respiratory tract (LRT) by transoral-tracheal lavage (TOTL), using qRT-PCR to determine the number of genome copies of SARS-CoV-2 present in the samples (Figure 4B,C). All the animals vaccinated with TNX-1800 showed no measurable viral RNA in URT six days post-challenge. The SARS-CoV-2 genome copy number in OP was observed below the limit of detection (<1000 copies/mL) in the group vaccinated by TNX-1800. In contrast significant genomic DNA was detected in all control NHPs with genome copies ranging from ~10^4^ to 10^6^ copies/mL (Figure 4B) (* *p* = 0.0285). A significant reduction of viral genome copies was observed in transoral-tracheal (TOTL) lavage by TNX-1800 compared to the diluent control (Figure 4C) (* *p* = 0.0328). The TNX-1800-vaccinated NHPs had viral loads below the limit of detection (<1000 copies/mL) in TOTL. In contrast, mock-vaccinated NHPs exhibited viral loads ranging from ~10^3^ to 10^4^ copies/mL at six days post-challenge. Altogether, the SARS-CoV-2 challenge brought about a 2-log to 4-log reduction of viral loads in the groups vaccinated with TNX-1800 in correlation with the groups vaccinated with the diluent.

### 3.2. Cynomolgus Monkeys (CMs) Challenge Study

#### Humoral Immune Responses in CMs

Similar to the AGM study, the TNX-1800 vaccination was well tolerated, and no adverse events were observed in CMs. Following vaccination, the “take” was observed via photographs of the vaccination site which were captured to measure lesion size. The mean lesion size in square centimeters, including the scar area that remained after the lesion had healed, is shown in Appendix A. Following vaccination, humoral responses were measured via ELISA and PRNT_50_ assays. The serum titers of SARS-CoV-2 spike antigen reactive IgG antibodies in all NHPs were measured at days 0 and 14 after vaccination and compared with pre-vaccination values (Figure 5A). In the TNX-1800-vaccinated group, endpoint SARS-CoV-2 spike protein specific isotype IgG1 titer means ranged between 95 and 5319 at days 0 and 14 post-vaccination, respectively. Neutralizing activity was measured at days 0, 14, and 29 post-vaccination by PRNT_50_ assay (Figure 5B). Fifty and seventy-five percent of the NHPs had neutralizing antibody titers at days 14 and 29 post-vaccination, respectively. The magnitude of the elicited neutralizing antibody response as measured by geometric mean titers (GMTs) detected 14- and 29-days post-vaccination ranged from 60 to 137.

### 3.3. Cell-Mediated Immune Responses

In addition to humoral responses, intracellular cytokine staining (ICS) by flow cytometry was also performed at days -174, -12, and 7 post-vaccination in all the CMs. The majority of the NHPs vaccinated with TNX-1800 had both CD4^+^ and CD8^+^ effector T-cell responses, as shown in Figure 6. The elevated levels of interferon-gamma (IFN-γ) expression at 7 days post vaccination were observed in both CD4^+^ (0.13%–0.43%) and CD8^+^ (0.04%–1.17%) T-cells expressed on day 7 post-vaccination in comparison to pre-vaccination (Figure 6A,C). However, both CD4^+^ and CD8^+^ T-cells showed no significant levels of changes in interleukin 4 (IL-4) levels at any time point (Figure 6B,D). 

### 3.4. Infectious Virus Load following SARS-CoV-2 Challenge

Viral loads in the upper and lower respiratory tracts after the SARS-CoV-2 challenge were assessed by Tissue Culture Infectious Dose 50 (TCID_50_) assay. At 2 days post-challenge, all mock-vaccinated NHPs had infectious virus in the BAL with a titer of ~5.0 log_10_ TCID_50_/mL. In contrast, the viral load in two NHPs in the TNX-1800-vaccinated group was at the limit of detection. Infectious virus was found only in two NHPs with viral loads 10 to 50-fold lower than the mock-vaccinated group, which ranged from 3.3–3.9 log_10_ TCID_50_/mL (Figure 7).

## 4. Discussion

As the global response to combat the SARS-CoV-2 pandemic continues, ongoing research and innovation are essential to developing additional safer, durable, and accessible vaccines [23]. While currently approved vaccines have played a significant role in saving lives, additional platforms, particularly ones that are easier or cheaper to manufacture and deliver, are needed to curtail this pandemic. The poxvirus vectors show great potential to address existing limitations and contribute to achieving global vaccination goals. Previously, we demonstrated the tolerability, safety, and immunogenicity of TNX-1800 in two animal models, Syrian hamsters and New Zealand white rabbits [21]. In the current report based on two NHP models, we confirm the immunogenicity and efficacy of TNX-1800, a live virus recombinant poxvirus vaccine candidate, against the SARS-CoV-2 challenge.

The work presented in this report demonstrates the vaccine immunogenicity and efficacy of TNX-1800 in two NHP species. TNX-1800 vaccination using the percutaneous method was well tolerated, with no reports of serious adverse events or significant changes in clinical parameters in the two NHP models, suggesting the possible use of alternative routes to deliver the vaccine effectively. A single dose of TNX-1800 administered to NHPs generated humoral responses. The neutralizing antibody responses induced by TNX-1800 were associated with upper and lower respiratory tract protection against mucosal challenge, meeting the primary immunogenicity and efficacy endpoints.

For CMs, a specific dose of TNX-1800 (6.1 log_10_ PFU) elicited a T-cell response, promoting both pathogen clearance and the generation of neutralizing antibody responses against SARS-CoV-2. The observed elevation of interferon-gamma (IFN-γ) levels in CMs vaccinated with TNX-1800 indicates a predominant TH1 immune response. IFN-γ is a pro-inflammatory cytokine crucial for antiviral defense, inhibiting viral replication and enhancing the activity of cytotoxic T cells and natural killer (NK) cells. This suggests that TNX-1800 effectively primed T cells to recognize SARS-CoV-2-infected cells and mounted a cellular immune response to clear the virus from the body.

The cytokine profiling assay demonstrated that TNX-1800 vaccination in CMs induced a TH1 response, promoted pathogen clearance, and generated neutralizing antibody responses against SARS-CoV-2. The vaccine’s ability to elicit a comprehensive immune response involving both antibody-mediated and cellular (T cell) immunity likely contributes to its effectiveness in promoting pathogen clearance and protection against SARS-CoV-2 infection. Additionally, the significant reduction of viral shedding in the upper airway and bronchoalveolar lavage indicates that TNX-1800 induces a protective mucosal immune response, which is critical in preventing infections at the entry sites of respiratory viruses and will play a role in reducing transmission.

Most licensed vaccines are administered by intramuscular or subcutaneous routes. These routes of vaccination do not elicit the sufficient mucosal IgG and IgA responses to provide durable protection against SAR-CoV-2-mediated disease. Although we have not formally tested neutralizing sIgA in the respiratory system after vaccination, the NHP data from oropharyngeal swabs, tracheal lavage (TL), and bronchoalveolar lavage (BAL) strongly suggest the presence of a humoral response at mucosal sites after a single vaccination. Currently, studies are being conducted to investigate the presence of spike-specific neutralizing antibodies at the mucosal sites following TNX-1800 vaccination.

Previously published data, in combination with our current results presented here, suggest that intradermal vaccinations generate mucosal T-cell memory and neutralizing sIgA responses. In addition, due to HPXV’s capacity to accommodate multiple antigens at various loci, rcHPXV-based vector vaccines may be able to elicit a more comprehensive immunity to targeted pathogens with a single vaccination. TNX-1800 can accommodate multiple SARS-CoV-2 genes such as Spike, Nucleocapsid protein (N), and the non-structural proteins (NSPs) associated with the RNA-dependent RNA Polymerase. These strategies warrant further investigation.

Given that the TNX-1800 vaccine can generate immune response with a single vaccination at small dose volumes, the vaccine is a viable candidate to be administered via microneedles. Further, the TNX-1800 vaccine is designed to enable single-dose, vial-sparing that can be manufactured using conventional cell culture systems for mass production, multi-dose packaging not dependent on ultra-cold-chain storage, and distribution. Future studies will compare the effect of different routes on mucosa and systemic immunity as well as efficacy against new variants of SARS-CoV-2.

## 5. Conclusions

The rcHPXV backbone platform, engineered as a monovalent vector expressing SARS-CoV-2 spike protein (TNX-1800), demonstrated the immunogenicity and efficacy following a single vaccination in two NHP models: AGMs and CMs. In both NHP species, TNX-1800 vaccination was well tolerated, indicated by the lack of serious adverse events or significant changes in clinical parameters. The results are consistent with the idea that just a few rounds of rcHPXV vector replication generated efficacious systemic immune responses against SARS-CoV-2 infection in the respiratory system. Taken together, these data strongly suggest that the HPXV can be developed into a modular “plug and play” multi-valent platform to provide efficacy against multiple pathogens.

## Figures and Tables

**Figure 1 vaccines-11-01682-f001:**
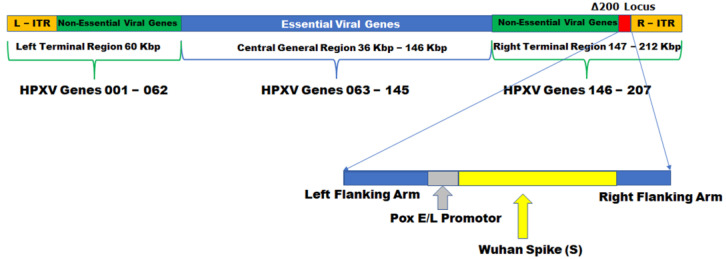
Schematic representation of the recombinant TNX-1800. TNX-1800 comprises the SARS-CoV-2 spike gene (Wuhan) (codon-optimized for VACV and under the control of a synthetic promotor) inserted into the Δ200 locus of the recombinant chimeric horsepox virus (rcHPXV). The resulting construct contains synthetic poxvirus early/late (Pox E/L) promoter-driven expression fragments for Spike glycoprotein armed with left and right HPXV flanking regions.

**Figure 2 vaccines-11-01682-f002:**
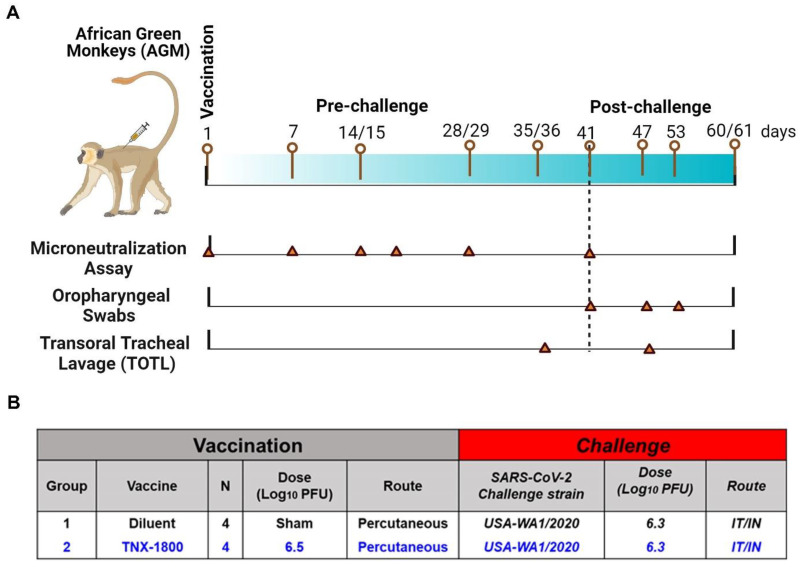
Study design for immunogenicity and efficacy of TNX-1800 in African Green Monkeys. (**A**) Study schedule for collecting serum samples from monkeys vaccinated with TNX-1800 and challenged with the SARS-CoV-2 isolate, USA-WA1/2020 on day 41 post-vaccination. (**B**) TNX-1800 dosing concentrations and dose administration routes adopted for pre- and post-challenge immunogenicity and efficacy study.

**Figure 3 vaccines-11-01682-f003:**
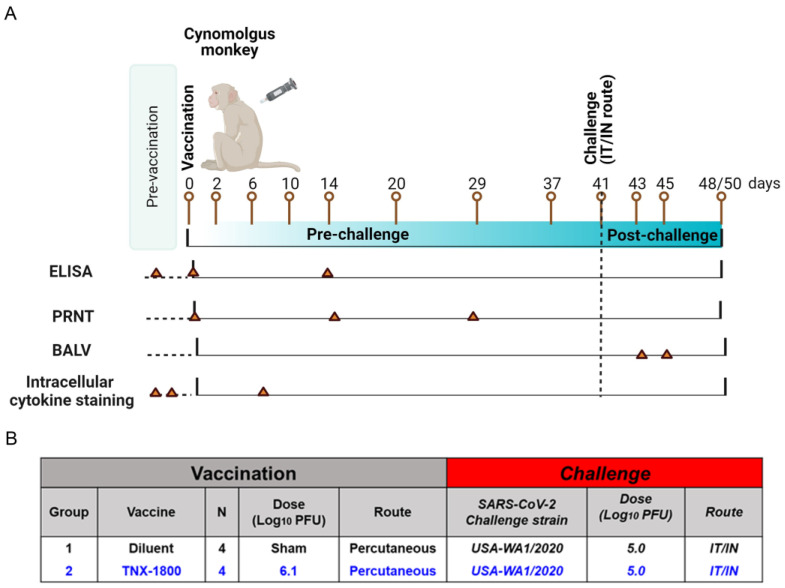
Study design for immunogenicity and efficacy of TNX-1800 in Cynomolgus monkeys. (**A**) Schedule for pre- and post-vaccination serum collection from monkeys vaccinated with TNX-1800 and challenged with the SARS-CoV-2 isolate, USA-WA1/2020 on day 41 post-vaccination. (**B**) TNX-1800 dosing concentrations and dose administration routes adopted for pre- and post-challenge immunogenicity and efficacy study in CM.

**Figure 4 vaccines-11-01682-f004:**
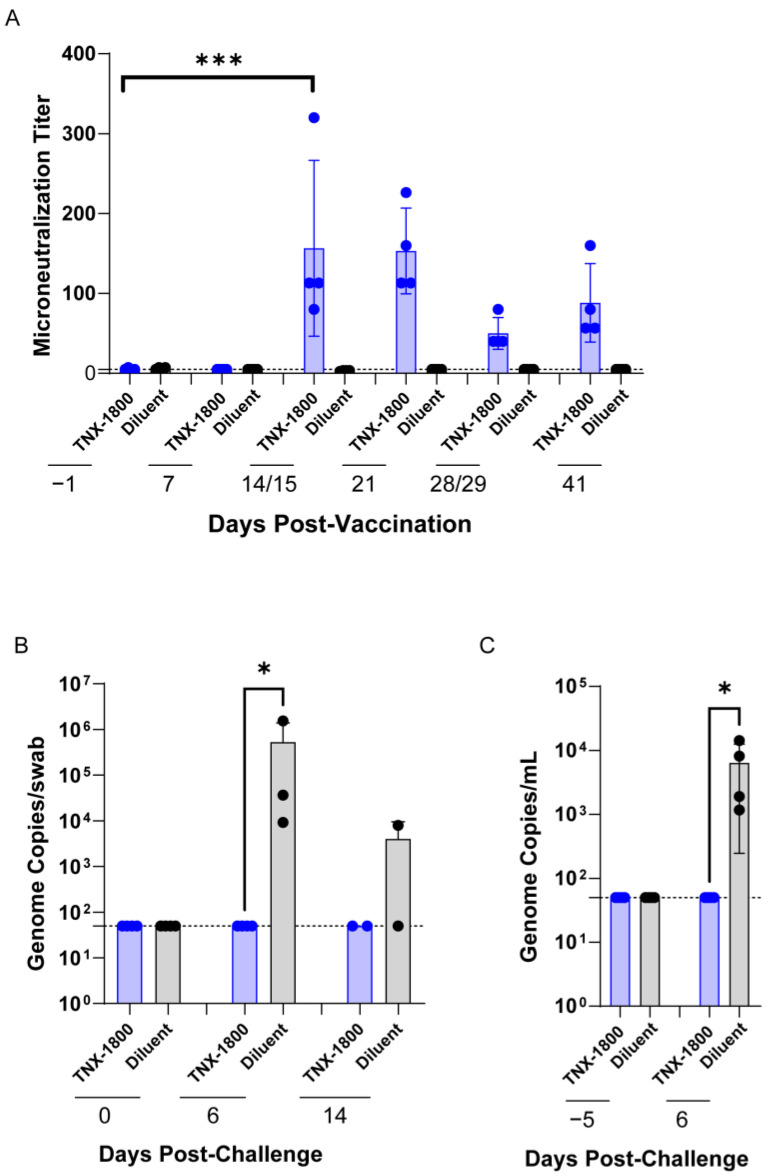
TNX-1800 immunogenicity and post-challenge viral shedding in African Green Monkeys. (**A**) Neutralizing antibody titers against SARS-CoV-2 were determined via microneutralization assay in serum samples collected at selected time points of the study (*** *p* = 0.0002). (**B**) SARS-CoV-2 load quantified in AGM oropharyngeal (OP) swabs by RT-qPCR. A significant reduction in SARS-CoV-2 shedding in oropharyngeal (OP) mucosa was observed in the group vaccinated by TNX-1800 compared to the control groups (* *p* = 0.0285). (**C**) SARS-CoV-2 load quantified in AGM tracheal lavage samples by RT-qPCR. A significant reduction of SARS-CoV2 shedding was observed in transoral-tracheal (TOTL) lavage by TNX-1800 (* *p* = 0.0328).

**Figure 5 vaccines-11-01682-f005:**
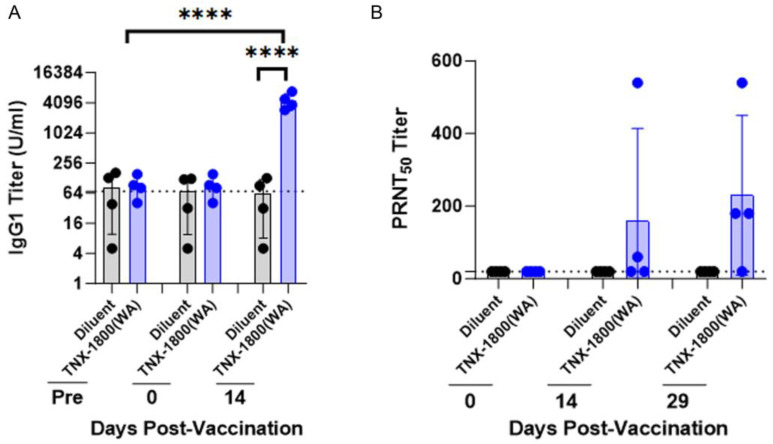
Total and neutralizing antibody titers against SARS-CoV-2 in TNX-1800-vaccinated Cynomolgus Monkeys. (**A**) Sera collected were analyzed in an ELISA for binding total IgG antibody levels to the SARS-CoV-2 spike antigen composed of three peptide fragments. The dilutions for the sera ranged from a reciprocal dilution of 50 to 6400 consisting of four-fold serial dilutions. The dashed lines indicate the limit of detection. Each data point depicts the individual animal, and the bar represents the group mean of the difference between OD450 and OD630 (**** *p* < 0.0001). (**B**) The sera from select time points outlined in Figure 2A were processed and analyzed for neutralizing antibody titers against the challenge virus, USA- WA1/2020. The dilutions for the sera ranged from a reciprocal dilution from 20 to 4860 consisting of three-fold dilutions. The dashed line indicates the limit of detection. Each data point depicts the individual animal.

**Figure 6 vaccines-11-01682-f006:**
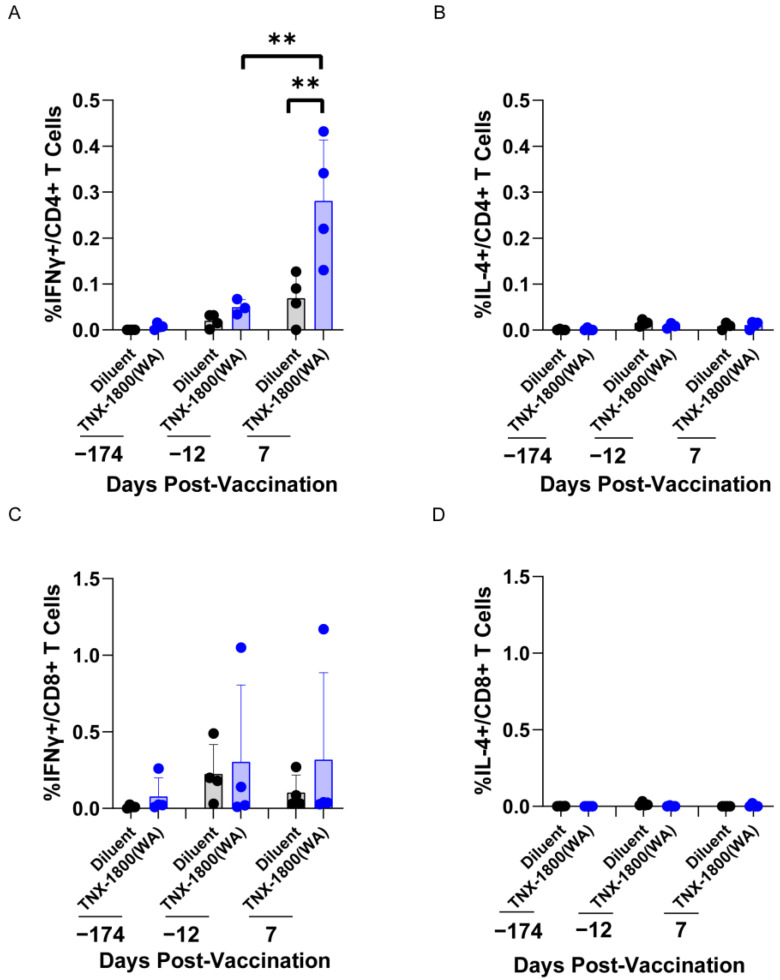
SARS-CoV2 spike specific cell-mediated immune response in TNX-1800 vaccinated Cynomolgus Monkeys. Percent distribution of Intracellular Cytokines in PMBCs is shown (**A**–**D**). PBMCs were collected, cryopreserved, and then intracellularly stimulated with spike antigens and stained with monoclonal antibodies against IL-4 and IFN-γ at selected time points. Significant elevation of IFN-γ expression was observed (** *p* = 0.0014).

**Figure 7 vaccines-11-01682-f007:**
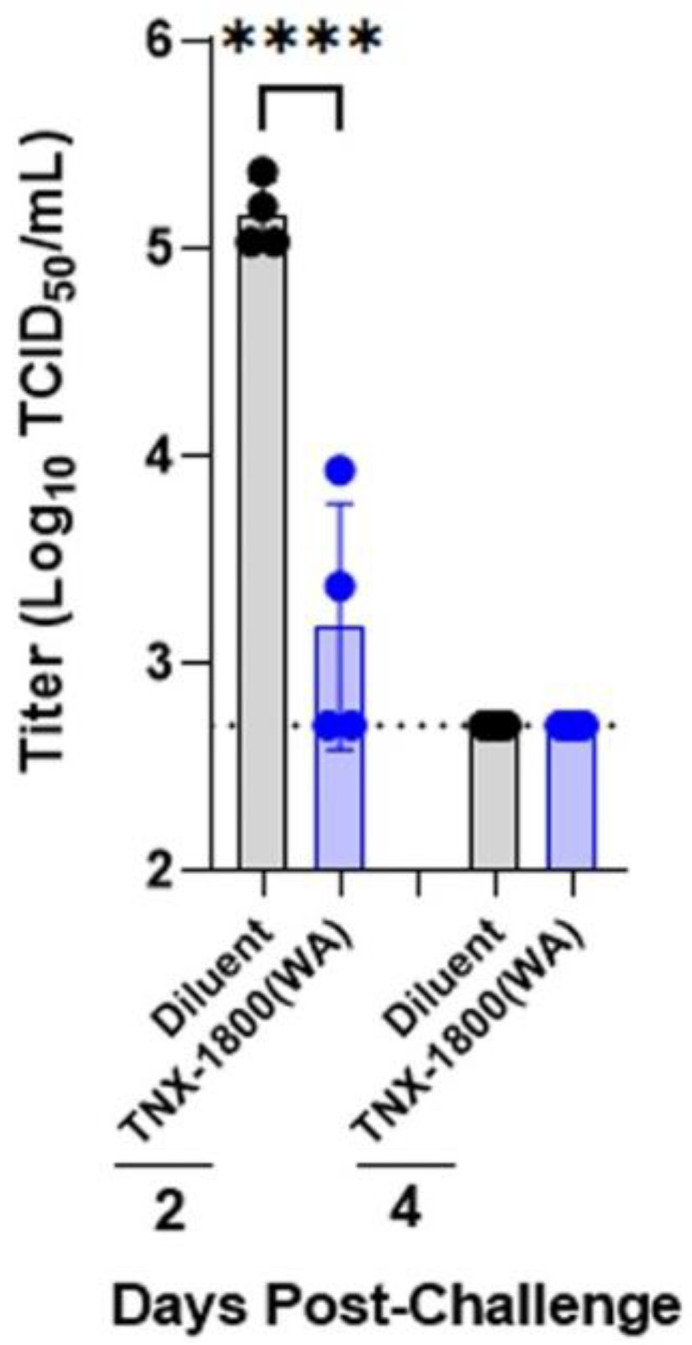
Infectious viral loads in BAL samples from SARS-CoV-2-challenged macaques. BAL samples from SARS-CoV-2-challenged monkeys were collected and tested for amounts of infectious virus loads by TCID_50_ assay. Each data point depicts the individual animal. A significant reduction (**** *p* < 0.0001) in infectious viral load was observed in the vaccinated group. The stippled line represents the lower limit of detection (2.70 log_10_ TCID_50_/mL).

## Data Availability

The data presented in this study are available on request from the corresponding author.

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
