# Peer review of "Immunogenicity and Efficacy of TNX-1800, A Live Virus Recombinant Poxvirus Vaccine Candidate, against SARS-CoV-2 Challenge in Nonhuman Primates"

_vaccines, 2023, doi:10.3390/vaccines11111682_

Round 1

Reviewer 1 Report

Comments and Suggestions for Authors

In this manuscript titled "Immunogenicity and Efficacy of TNX-1800, A Live Virus Recombinant Poxvirus Vaccine Candidate, Against SARS-CoV-2 Challenge in Nonhuman Primates", the authors tested a single dose vaccination of TNX-1800 in two NHP species and found a reasonable antibody response after vaccination and a clear reduction of viral load after SARS-CoV-2 challenge. Overall, this is a relatively interesting study. However, the study is not well designed, specifically, sample size is too small, no other vaccination formula was compared, immune responses were only examined after vaccination but not after challenge. Due to the difficulty of carrying out NHP studies, it may be hard for authors to repeat this study with a better design in a reasonable time. Therefore, I would recommend a major revision of the text to discuss and acknowledge the caveats.

Major issue:

1. As summarized above, please include a discussion of this study's caveats in the discussion section.

2. Line 48 "... and protein subunit vaccines from [9], and Novavax [10]", this sentence is incomplete, please revise.

3. Line 413-414 "Remarkably, a single dose of TNX-1800 administered to NHPs resulted in robust humoral responses." I can see humoral responses were generated by this vaccination. However, I didn't see any control or comparison to other vaccination formula. So I can't conclude on whether the humoral responses are "remarkable" and "robust". The expression here is an exaggeration, please revise the language.

4. Line 428 "This balanced immune response is essential..." The authors didn't see any IL4 mediated T cell response, how come the authors concluded that the immune response was balanced? In addition, Tfh cells could secrete IL-4 too. Please re-write the whole paragraph about Th1/Th2 response, or delete it.

5. Line 448 "Although we have not formally tested neutralizing sIgA in the respiratory system after vaccination..." This paragraph is extremely confusing. The authors neither tested IgA in this study, nor compared different vaccination routes, yet the authors spent a whole paragraph to discuss these two points. Please delete this whole paragraph because they are irrelevant to the data shown in this study.

Minor issue:

Line 55-59 is repetitive, please revise to make it more concise.

Author Response

Comment #1: In this manuscript titled "Immunogenicity and Efficacy of TNX-1800, A Live Virus Recombinant Poxvirus Vaccine Candidate, Against SARS-CoV-2 Challenge in Nonhuman Primates", the authors tested a single dose vaccination of TNX-1800 in two NHP species and found a reasonable antibody response after vaccination and a clear reduction of viral load after SARS-CoV-2 challenge. Overall, this is a relatively interesting study. However, the study is not well designed, specifically, sample size is too small, no other vaccination formula was compared, immune responses were only examined after vaccination but not after challenge. Due to the difficulty of carrying out NHP studies, it may be hard for authors to repeat this study with a better design in a reasonable time. Therefore, I would recommend a major revision of the text to discuss and acknowledge the caveats.

Response #1 Response #1: All the authors would like to thank the reviewer for valuable suggestions and feedback.

Comment #2: As summarized above, please include a discussion of this study's caveats in the discussion section.

Response #2: The authors thank the reviewer for the insightful comments. We respectfully disagree with the reviewer on the fact that the study is not well designed. These are expensive “proof-of-concept” studies that were performed at the height of the COVID-19 pandemic when both non-human primate species were limited in availability and limited high containment laboratories were available that could perform these experiments. The Cynomolgus macaques’ study itself was almost a million dollars study. For these reasons, we designed a study with 80% power to detect vaccine efficacy. However, future studies will compare the effect of different routes on mucosa and systemic immunity as well as efficacy against new variants of SARS-CoV-2. Please refer to lines 454-455.

Comment #3:  Line 48 "... and protein subunit vaccines from [9], and Novavax [10]", this sentence is incomplete, please revise.

Response #3: We thank the reviewer for pointing out this error. We have corrected it in the revised manuscript. Please refer to lines 47-48.

Comment #4:   Line 413-414 "Remarkably, a single dose of TNX-1800 administered to NHPs resulted in robust humoral responses." I can see humoral responses were generated by this vaccination. However, I didn't see any control or comparison to other vaccination formula. So I can't conclude on whether the humoral responses are "remarkable" and "robust". The expression here is an exaggeration, please revise the language.

Response #4: We thank the reviewer for the careful and insightful review of our manuscript. We have revised the manuscript as suggested. Please refer to lines 416.

Comment #5:   Line 428 "This balanced immune response is essential..." The authors didn't see any IL4-mediated T-cell response, how come the authors concluded that the immune response was balanced? In addition, Tfh cells could secrete IL-4 too. Please re-write the whole paragraph about Th1/Th2 response, or delete it.

Response #5: We agree with the reviewer and thank the reviewer for pointing this out. We have corrected it in the revised manuscript.

Comment #6:   Line 448 "Although we have not formally tested neutralizing sIgA in the respiratory system after vaccination..." This paragraph is extremely confusing. The authors neither tested IgA in this study, nor compared different vaccination routes, yet the authors spent a whole paragraph to discuss these two points. Please delete this whole paragraph because they are irrelevant to the data shown in this study.

Response #6: We thank the reviewer and agree with the comment. These studies are currently being conducted. However, due to the lack of viral shedding, the data suggests mucosal immunity in the upper respiratory tract. We have revised the paragraph to make it more concise. Please refer to lines 437-440.

Comment #7:   Line 55-59 is repetitive, please revise to make it more concise.

Response #7: We thank the reviewer for the critical review of our manuscript. We have revised the sentences as suggested. Please refer to lines 54-58.

Reviewer 2 Report

Comments and Suggestions for Authors

This valuable article describes the use of a recombinant chimeric horsepox virus (rcHPXV) (TNX-801) as a vaccine platform against monkeypox virus (MPXV). This vector was previously used to immunize NHPs and protected against lethal MPXV clade I challenge in NHPs.

Then TNX-801 platform was engineered to express SARS-CoV-2 spike protein of the Wuhan strain (TNX-1800). After showing safety, immunogenicity and protective capacity in NZ rabbits and Syrian golden hamsters respectively they are now proposing the present proof-of-concept report, describing the immunogenicity and efficacy of TNX-1800 in nonhuman primates, African Green Monkeys (AGMs) and Cynomolgus Macaques (CMs) using a single dose of TNX-1800 vaccination followed by a SARS CoV-2 challenge, the vaccine was able to significantly reduce virus replication/shedding in the respiratory track.

This reviewer considers that this document has a group of deficiencies that must be overcome prior to publication:

1- I disagree that the platform is promising in the global effort to stop the spread of the pandemic. This phrase is already extemporaneous. The article does not provide information on neutralizing activity on new variants of SARS-CoV-2 and leaves this for the future when in reality it would be very informative given the antibody response that is generated and it is also easy to carry out. Currently, the potential application of a vaccine is as a booster and in the case of the application of live vaccines the cost-benefit balance is not promising. Even accepting the existence of gaps in the new vaccine messenger RNA technology. Maybe the conclusions should focus on the real contributions of the article to prevent transmission etc.

2- This reviewer disagree with the authors' assessment of the existence of a strong cellular response. We must consider that there are no significant increases in the frequency of CD8+ cells secreting gamma IFN, leaving this response reduced only to that obtained at the level of frequency of CD4+ cells secreting gamma interferon. There are other vaccines designed for a dose where the response levels are more balanced and much more powerful (https://www.nature.com/articles/s41586-020-3035-9). The use of this type of terms must be discussed / substantiated or compared to other results of the state of the art.

3- I find that there are consistency problems: the high levels of antibodies are not consistent with the poor IL4 response and the Th1 pattern that is suggested as a result of detecting only the otherwise poor IFN gamma response. This aspect should be discussed and maybe the experimental result repeated if the result has not been reproduced.

4- The document has limitations from a formal point of view: the description of the statistics is very poor, it should be made more substantial, explaining the statistical methods used in each experiment. This deficiency may affect the analysis of the results. The document must be reviewed in a general way to correct spelling errors, for example in 2.10 part of the first word is omitted.

5- It is surprising to observe that in the final paragraph, when the aim is to conclude or summarize the work, the authors begin by citing reference 20 (again), this paragraph should focus on the real conclusions of this work, linked to the results obtained that undoubtedly have their value.

Comments on the Quality of English Language

The document must be reviewed in a general way to correct spelling errors, for example in 2.10 part of the first word is omitted.

Author Response

Comment #1: This valuable article describes the use of a recombinant chimeric horsepox virus (rcHPXV) (TNX-801) as a vaccine platform against monkeypox virus (MPXV). This vector was previously used to immunize NHPs and protected against lethal MPXV clade I challenge in NHPs.

Then TNX-801 platform was engineered to express SARS-CoV-2 spike protein of the Wuhan strain (TNX-1800). After showing safety, immunogenicity and protective capacity in NZ rabbits and Syrian golden hamsters respectively they are now proposing the present proof-of-concept report, describing the immunogenicity and efficacy of TNX-1800 in nonhuman primates, African Green Monkeys (AGMs) and Cynomolgus Macaques (CMs) using a single dose of TNX-1800 vaccination followed by a SARS CoV-2 challenge, the vaccine was able to significantly reduce virus replication/shedding in the respiratory track.

This reviewer considers that this document has a group of deficiencies that must be overcome prior to publication:

Response #1: All the authors would like to thank the reviewer for valuable suggestions and feedback.

Comment #2:  I disagree that the platform is promising in the global effort to stop the spread of the pandemic. This phrase is already extemporaneous. The article does not provide information on neutralizing activity on new variants of SARS-CoV-2 and leaves this for the future when in reality it would be very informative given the antibody response that is generated and it is also easy to carry out. Currently, the potential application of a vaccine is as a booster and in the case of the application of live vaccines the cost-benefit balance is not promising. Even accepting the existence of gaps in the new vaccine messenger RNA technology. Maybe the conclusions should focus on the real contributions of the article to prevent transmission etc.

Response #2: The authors thank the reviewer for his/her insightful comments. We agree with the reviewer and have added the conclusion section focusing on the real contribution of the study. Please refer to lines 462-470. We also understand the need for this suggested additional data on neutralizing activity on new variants of SARS-CoV-2 to make this work more significant in the field. We are currently performing this study but since the work is in progress and needs more confirmatory experiments it would be early to include the preliminary findings in this study. Please refer to lines 459-460.

Comment #3: This reviewer disagree with the authors' assessment of the existence of a strong cellular response. We must consider that there are no significant increases in the frequency of CD8+ cells secreting gamma IFN, leaving this response reduced only to that obtained at the level of frequency of CD4+ cells secreting gamma interferon. There are other vaccines designed for a dose where the response levels are more balanced and much more powerful (https://www.nature.com/articles/s41586-020-3035-9). The use of this type of terms must be discussed / substantiated or compared to other results of the state of the art.

Response #2: We agree with the comment and thank the reviewer for pointing it out. We have revised the sentences accordingly in the revised manuscript. Please refer to lines 27, 420, 429.

Comment #3:  I find that there are consistency problems: the high levels of antibodies are not consistent with the poor IL4 response and the Th1 pattern that is suggested as a result of detecting only the otherwise poor IFN gamma response. This aspect should be discussed and maybe the experimental result repeated if the result has not been reproduced.

Response #3: We thank the reviewer for pointing out the use of exaggerated sentences to describe the Th1 response. We agree with the reviewer, and we have revised the sentences accordingly. Please refer to the lines 424-427.

Comment #4:  The document has limitations from a formal point of view: the description of the statistics is very poor, it should be made more substantial, explaining the statistical methods used in each experiment. This deficiency may affect the analysis of the results. The document must be reviewed in a general way to correct spelling errors, for example in 2.10 part of the first word is omitted.

Response #4: The authors thank the reviewer for this suggestion. We apologize for that and have revised the method section for statistical analysis and have reviewed the manuscript thoroughly for the spell checks. Please refer to lines 304-308.

Comment #5:  It is surprising to observe that in the final paragraph, when the aim is to conclude or summarize the work, the authors begin by citing reference 20 (again), this paragraph should focus on the real conclusions of this work, linked to the results obtained that undoubtedly have their value.

Response #5: We thank the reviewer for the critical review. We have revised the final paragraph for clarity and have included the conclusion section focused on the real contribution of the study in the revised manuscript. Please refer to lines 459-470.

Reviewer 3 Report

Comments and Suggestions for Authors

Introduction and design

This is a meticulously planned and documented study with clear graphics.

Analysis

The analysis is thorough and complete.

Your tests involved eight African Green Monkeys and eight Cynomolgus Macaques. Please provide data showing the range of response for key outcome measures across the sixteen individuals. This is important to demonstrate howe far data from groups of this size can be generalized.

Conclusions

I realise that the conclusion section can be used for definite conclusions and also for reasonable speculations to guide further research.

You wrote: ”Most licensed vaccines are administered by intramuscular or subcutaneous routes. These mechanisms of vaccination may not appropriately prime the mucosal lymph nodes or produce sufficient IgA responses to provide durable protection against SAR-CoV-2 mediated disease. In contrast, intradermal vaccination has the potential to generate respiratory, secretory IgA (sIgA) and memory mucosal T-cell responses.”

[You make some important and interesting assertions here using the words “may not“ and “has the potential.” Can you provide numerical comparisons to support you assertions please?]

You made two further assertions:

”Previously published data, in combination with our results strongly suggest that intradermal vaccinations generate long-lasting mucosal T-cell memory and neutralizing sIgA responses. In addition, due to HPXV’s capacity to accommodate multiple antigens at various loci, HPXV-based vector vaccines may be able to elicit a more comprehensive immunity to targeted pathogens with a single vaccination.’

[can you please provide numerical data to support your assertions “strongly suggest” and “may be able”]

Another important assertion: “TNX-1800 has the potential to accommodate multiple SARS-CoV-2 genes such as Spike, Nucleocapsid protein (N), and the non-structural proteins (NSP) associated with the RNA-dependent RNA Polymerase.”

[this will of great interest to other researchers. You write “has the potential.” Can you provide numerical data please?]

Minor typos and clarifications:

48  from [9],   [missing citation]

on cold chain. [change to on the ]

57  Vaccina vi- [change to Vaccinia]

61 mouse, Syrian hamsters, and non-human primates’ [change to primates. You do not need the ’]

80 (ATCC). [delete the .]

via the intranasal/intratracheal (IN/IT) route [both routes?]

121 naïve monkeys [naïve to?]

123 adult monkeys [also naïve to?]

[you do clarify this later at 158 “The animals were challenged via the intranasal route and followed by the intratracheal route”]

294 pipettor. [pipette?]

Author Response

Comment #1:  This is a meticulously planned and documented study with clear graphics. Analysis: The analysis is thorough and complete.

Response #1: All the authors would like to thank the reviewer for leaving us feedback.

Comment #2:  Your tests involved eight African Green Monkeys and eight Cynomolgus Macaques. Please provide data showing the range of response for key outcome measures across the sixteen individuals. This is important to demonstrate howe far data from groups of this size can be generalized.

Response #2: We thank the reviewer for the careful and insightful review of our manuscript. We have updated it in the revised manuscript. For the data corresponding to the range of responses please refer to lines 318, 327, 331, 354, 359, 376, 393.

Comment #3:  Conclusions: I realise that the conclusion section can be used for definite conclusions and also for reasonable speculations to guide further research.

Response #3: We thank the reviewer for the constructive comment. We have included the conclusion section in the revised manuscript. Please refer to line 460.

Comment #4:  You wrote: ”Most licensed vaccines are administered by intramuscular or subcutaneous routes. These mechanisms of vaccination may not appropriately prime the mucosal lymph nodes or produce sufficient IgA responses to provide durable protection against SAR-CoV-2 mediated disease. In contrast, intradermal vaccination has the potential to generate respiratory, secretory IgA (sIgA) and memory mucosal T-cell responses.”

[You make some important and interesting assertions here using the words “may not“ and “has the potential.” Can you provide numerical comparisons to support you assertions please?]

Response #4: We thank the reviewer for pointing it out. We have revised these assertive sentences in the revised manuscript.  Please refer to the lines 437-444

Comment #5:  You made two further assertions:

”Previously published data, in combination with our results strongly suggest that intradermal vaccinations generate long-lasting mucosal T-cell memory and neutralizing sIgA responses. In addition, due to HPXV’s capacity to accommodate multiple antigens at various loci, HPXV-based vector vaccines may be able to elicit a more comprehensive immunity to targeted pathogens with a single vaccination.’

[can you please provide numerical data to support your assertions “strongly suggest” and “may be able”]

Response #5: We thank the reviewer for pointing out this issue. We have revised these assertive sentences in the revised manuscript.  Please refer to lines 445-447

Comment #6:  Another important assertion: “TNX-1800 has the potential to accommodate multiple SARS-CoV-2 genes such as Spike, Nucleocapsid protein (N), and the non-structural proteins (NSP) associated with the RNA-dependent RNA Polymerase.”

[this will of great interest to other researchers. You write “has the potential.” Can you provide numerical data please?]

Response #6: We thank the reviewer for their helpful comments. We have identified >5 loci in the HPXV genome containing ~200 ORFs which could accommodate single or multiple transgenes. New constructs with different transgenes of SARS-CoV-2 are under development and will be studied in the future. We have also restructured the assertive sentences in the revised manuscript.  Please refer to line 450.

Comment #7:  48 from [9],   [missing citation]

Response #7: We thank the reviewer for pointing out this error. We have corrected it in the revised manuscript. Please refer to lines 47-48.

Comment #8:  on cold chain. [change to on the]

Response #8: We thank the reviewer for pointing out this error. We have corrected it in the revised manuscript. Please refer to line 53.

Comment #9:  57 Vaccina vi- [change to Vaccinia]

61 mouse, Syrian hamsters, and non-human primates’ [change to primates. You do not need the ’]

Response #9: We thank the reviewer for pointing out this error. We have corrected it in the revised manuscript. Please refer to line 57, line 59.

Comment #10:  80 (ATCC). [delete the .]

Response #10: We thank the reviewer for pointing out this. However, “.” is at the end of a sentence.  Please refer to line 76.

Comment #11:  via the intranasal/intratracheal (IN/IT) route [both routes?]

Response #11: We thank the reviewer for pointing out this error. We have corrected it in the revised manuscript. Please refer to line 116.

Comment #12:  121 naïve monkeys [naïve to?]

Response #12: We thank the reviewer for this comment. These were juvenile NHPs that were not used in any other study previously.

Comment #13:  123 adult monkeys [also naïve to?]

[you do clarify this later at 158 “The animals were challenged via the intranasal route and followed by the intratracheal route”]

Response #13: We thank the reviewer for this comment. These were juvenile NHPs that were not used in any other study previously.

Comment #14:  294 pipettor. [pipette?]

Response #14: We thank the reviewer for pointing out this error. We have corrected it in the revised manuscript. Please refer to the line 291

Round 2

Reviewer 1 Report

Comments and Suggestions for Authors

In author's response #2, " we designed a study with 80% power to detect vaccine efficacy". This is actually an important point, but I didn't see this point being described in the manuscript. Please include how authors used statistics to determine the sufficient number of NHP to perform these two studies in the method section.

Regarding the author's response #3, please double check the grammar of this sentence again.

All other concerns resolved.

Author Response

Comment 1: In author's response #2, " we designed a study with 80% power to detect vaccine efficacy". This is actually an important point, but I didn't see this point being described in the manuscript. Please include how authors used statistics to determine the sufficient number of NHP to perform these two studies in the method section.

Response #1: We thank the reviewer for this comment. We have included the suggested information. Please refer to lines 304-308. 

Comment 2: Regarding the author's response #3, please double check the grammar of this sentence again.

Response #2: We thank the reviewer for the critical review of our manuscript. We have revised the sentence as suggested. Please refer to lines 47-48. 

Reviewer 2 Report

Comments and Suggestions for Authors

Fine, please just check sentence in line 386 and complete the idea.

Author Response

Comment 1: Fine, please just check sentence in line 386 and complete the idea.

Response 1: We thank the reviewer for this comment. We have revised the sentence as suggested. Please refer to line 388. 

Reviewer 3 Report

Comments and Suggestions for Authors

Thank you for the meticulous corrections 

Author Response

Comment: Thank you for the meticulous corrections 

Response: We thank the reviewer for reviewing our manuscript. It has certainly improved the overall quality of the manuscript.